# Pitfalls in Evaluating Inference-time Methods for Improving LLM Reliability

**Michael Jerge**                                                                    *mj6ux@virginia.edu*
*Department of Computer Science*
*University of Virginia*

**David Evans**                                                                        *evans@virginia.edu*
*Department of Computer Science*
*University of Virginia*

**Reviewed on OpenReview:** *https://openreview.net/forum?id=xeGWsmqFS8*

## Abstract

Large Language Models (LLMs) have demonstrated remarkable capabilities but are still prone to outputting falsehoods using seemingly persuasive language. Many recent works attempt to address this problem by using LLMs in a framework where a single seed prompt results in a series of interactions involving augmented prompts with an otherwise unchanged LLM, and the results are aggregated with a goal of producing a more reliable output. We consider the replicability and generalizability of evaluations of inference-time methods intended to improve the reliability of responses from base LLMs. We survey how methods have been evaluated in the literature and find a great variety of benchmarks and models in use. Motivated by this, we conduct our own evaluation to evaluate the effectiveness of a few methods across a range of benchmarks and models. We find that while these techniques show promise in improving reliability, there is still significant variability in performance across different domains and tasks, and methods that show substantial improvements on weaker base models often do not improve reliability for better base models.

## 1 Introduction

Large language models (LLMs) have made remarkable progress but are still unable to reliably provide factual responses. Research on LLM reliability has increased with the widespread excitement about these models and recognition of their current limitations. Methods aiming to improve reliability have been proposed across various stages of the LLM lifecycle, including training, deployment, and inference. Training methods include various methods for fine-tuning and knowledge distillation. Reliability can also be improved by incorporating methods in how an LLM is deployed, such as retrieval-augmented generation and integration with external tools and knowledge sources. All of these types of methods can contribute to improving LLM reliability and are worth investigating. In this study we limit our focus to methods that are intended to improve the reliability of an underlying LLM by changing the way it is used at inference time. Although there is often value in considering the impact of changes in a full system with a particular LLM and on a specific task, many works claim to provide general methods for improving reliability at inference time, with expectations that performance improvements in tested settings generalize to other tasks and different base models. The goal of our work is to understand the robustness and generalizability of inference-time LLM methods.

We focus on approaches that are implemented within the generalized pipeline where a user submits a seed prompt that is intended to capture what the user wants from the LLM system, the system performs some automated computations including interactions with an underlying LLM to generate a response to that seed prompt, and then a response is presented to the user. We explore system-level inference-time methods intended to improve reliability limiting our study to methods that do not rely on any additional training

or tuning, or on external knowledge sources or tools. These comprise a limited, but important, part of the solution space being explored to improve LLM reliability. Many of these methods involve redundancy and variation, combining multiple interactions with an underlying LLM with the goal of deriving a better response than the one that would be obtained from a single, straightforward submission of the prompt. These methods are complementary with methods to improve underlying models, but differ in that they can readily be applied to any LLM.

**Contributions.** We survey the evaluation of inference-time methods to improve LLM reliability. We analyze how research in this area evaluates the effectiveness of proposed methods (Section 3). We find a broad range of approaches with little consensus on the benchmarks (Section 3.2) and models (Section 3.3) that are used in these evaluations. Motivated by these findings, we conduct an experiment to comprehensively evaluate a few methods with a group of representative benchmarks and a diverse set of models (Section 4). Our main findings are that claims about the effectiveness of inference-time LLM methods are fragile to both the models and benchmarks used. In particular, methods that result in large improvements for weaker models and on poorly-chosen benchmarks often have disappointing performance when evaluated more comprehensively. Lastly, we provide recommendations to researchers on ways to improve LLM reliability studies.

## 2   Related Work

Many works have conducted evaluations on the reliability of LLMs. Notable examples include Chang et al. (2024) and the HELM project (Liang et al., 2023). Other works have focused on evaluating the reliability of a single LLM. For example, Shen et al. (2023) studied ChatGPT. Yang et al. (2024) focused on analyzing the performance of LLMs on specific downstream tasks. Wang et al. (2023a) evaluate particular characteristics of specific models and examined various aspects of trustworthiness, including toxicity, bias, robustness, privacy, ethics, and fairness on GPT-3.5-turbo and GPT-4. Kadavath et al. (2022) studied reliability of LLM responses and whether it is possible to predict whether a model's response is reliable. These works (and many others) share with ours the broader goal of understanding the performance of LLMs, but all of them focus on the unaided model, whereas our focus is on inference-time methods that are intended to improve the reliability of an underlying model.

Prompt engineering, defined by Reynolds & McDonell (2021) as methods whereby humans iteratively modify prompts to elicit desired behaviors from LLMs, is also a common technique used to improve LLM outputs, and there is extensive work in this area. Schulhoff et al. (2024) provide a comprehensive taxonomy of prompt engineering techniques, many of which can be used in conjunction with the inference-time methods that we study. Similarly, answer engineering involves crafting or selecting algorithms to extract precise answers from LLM outputs, often requiring human involvement (Schulhoff et al., 2024, p. 17). Although some notions of prompt engineering are broad enough to include many of the inference-time methods we consider here, prompt engineering typically involves manual human effort (at either the task or individual query level) and post-hoc refinement in selecting seed prompts, which is outside the scope of the general-purpose and automated inference-time methods we consider here.

Mialon et al. (2023) surveys LLMs that are augmented with reasoning capabilities and external tools and considers how to evaluate these augmented LLMs. They include some inference-time methods like the ones we focus on in this paper, but the main emphasis is on the use of external tools like web search engines and symbolic reasoning modules. Welleck et al. (2024) reviews different types of inference-time algorithms used with LLMs. Although we focus on methods that work with any LLM without modification and his study includes methods requiring additional training or external tools, there is some overlap between the methods evaluated. Our evaluation of these methods across various benchmarks and models, including state-of-the-art LLMs, provides novel insights and recommendations for future reliability studies. Chu et al. (2024) provides a comprehensive taxonomy of reasoning methods, with a focus on categorizing Chain of Thought prompting techniques, but does not examine evaluation practices or reproducibility concerns. While their work focuses on organizing and categorizing different reasoning approaches, our study empirically tests whether performance claims about these methods are robust across different models and benchmarks.

The work closest to ours is the review by Sprague et al. (2024), focused on the popular Chain of Thought method (Section 4.1). Their approach of evaluating methods with common goals across a standard set of datasets and benchmarks is similar to what we do in Section 4, and several of our conclusions are consistent with their results. In particular, the authors found that advantages of using the Chain of Thought method are largely limited to improving symbolic execution, although it does not perform as well as external symbolic solvers. Sprague et al. (2024) is limited to studying just the Chain of Thought method, whereas our study considers a range of inference-time methods (including Chain of Thought) for LLM reliability. Our work combines an analysis of evaluation practices in the literature with an empirical study designed to highlight pitfalls in current evaluation methodologies.

## 3 Evaluations in the Literature

To understand how inference-time reliability methods are evaluated in the research literature, we conducted a comprehensive analysis, aided by an automated tool, of the evaluation approaches employed in the relevant research literature (Section 3.1). For each paper, we catalog the evaluation benchmarks and models used by the authors to test their proposed methods. Our analysis reveals a large variety of different benchmarks (Section 3.2) and models (Section 3.3) in use; no single model or benchmark is used in more than a third of the papers. This motivates our experiments in Section 4 to understand how robust evaluations are to choices of underlying models and benchmarks.

### 3.1 Literature selection

To identify relevant papers on inference-time methods for improving LLM reliability, we implemented an automated system using the Semantic Scholar API.[1] We selected the Chain of Thought paper (Wei et al., 2022) as our starting point due to its significantly higher citation count compared to other foundational inference-time methods. Appendix A summarizes a citation overlap analysis based on alternative starting papers that supports this choice as capturing both a substantial core of relevant work and papers that would not be found using different starting points. This starting point resulted in an initial set of 6 318 papers.

To enable automated analysis and avoid licensing issues, we then filtered this set to only papers available on arXiv, which reduced the set to 4 895 papers. Of these, our automated script successfully retrieved 4 886 papers; the nine unsuccessful retrievals were all papers that had been withdrawn from arXiv. Our dataset spans from January 2022, when Wei et al. (2022) was posted on arXiv, through the end of 2024 (our collection search was run on 7 January 2025). We retrieved the full text of each paper using the arXiv API, implementing appropriate rate limiting and error handling to ensure reliable data collection.

To validate the automated analysis, we manually analyzed a set of 50 papers to extract model and benchmark data. For these 50 papers, we compared the results from the manual analysis to those from the automated GPT-4o analysis, finding a Jaccard similarity index of 0.887 for benchmark categorization and 0.874 for model categorization. Our examination of the discrepancies revealed three main types. First, there were four instances where the automated analysis was more precise than human annotation, such as when the human analysts incorrectly classified certain models as foundational in papers like "ReAct" (Yao et al., 2023d). We found five cases where ambiguous terminology in the original papers led to different but equally justifiable interpretations between human and automated analysis, as seen in papers discussing variations of benchmark names. There were also occasional mistakes by the automated system, such as missing the text-curie-001 model in the "Boosted Prompt Ensembles" paper (Pitis et al., 2023).

The automated analysis showed consistency in identifying both common and rare combinations of models and benchmarks, suggesting reliable performance across the full spectrum of papers. These findings suggest that the high similarity index and the nature of the discrepancies support that the data resulting from our automated analysis is high enough quality to use for our purposes. Additional notes and inconsistencies between the categorizations can be found in the literature_analysis subdirectory in our repository.

---

[1]The code for our automated system and all of the data and code to reproduce our work, is available under an open source license in this public repository: https://github.com/mmjerge/LLM-Evaluation-Framework.

Table 1: Twenty most frequently used benchmarks and models in evaluations (Jan 2022–Dec 2024).

| Benchmarks | | Models | |
|---|---|---|---|
| **Benchmark** | **Number of Papers** | **Model** | **Number of Papers** |
| GSM8K | 425 | GPT-4 | 1835 |
| MATH | 174 | GPT-3.5-Turbo | 1739 |
| MMLU | 171 | GPT-3 | 705 |
| SVAMP | 139 | LLaMA-2-7B | 485 |
| StrategyQA | 126 | PaLM-2 | 415 |
| HotpotQA | 120 | LLaMA-3-8B | 376 |
| HumanEval | 119 | GPT-4o | 374 |
| TruthfulQA | 81 | Mistral-7B | 343 |
| CommonsenseQA | 77 | LLaMA-2-70B | 282 |
| HellaSwag | 70 | LLaMA-2-13B | 270 |
| AQuA | 64 | BERT | 263 |
| TriviaQA | 63 | LLaMA | 229 |
| MBPP | 63 | LLaMA-7B | 203 |
| MultiArith | 61 | LLaMA-70B | 188 |
| Winogrande | 56 | RoBERTa | 159 |
| BoolQ | 55 | GPT-4V | 148 |
| PIQA | 52 | T5 | 134 |
| OpenBookQA | 49 | Vicuna-13B | 125 |
| ASDiv | 46 | LLaMA-2 | 119 |
| SQuAD | 44 | GPT-2 | 107 |

Following this assurance mechanism, we used our automated system to systematically extract and categorize information from our corpus. This automated extraction was designed with specific criteria, though we encountered technical limitations when processing certain papers due to tokenization conflicts with special tokens like '<|endoftext|>' in the source text. Thus, we were unable to catalog requisite information for twelve of the papers. For benchmarks, we included both standard evaluation datasets and custom evaluation sets, while excluding datasets used solely for training. For models, we captured all baseline comparisons, proposed models, and variants tested in ablation studies, but excluded models that were only referenced without experimental evaluation. We canonicalize benchmark and model names following the categorizations, as the same model or benchmark could have been classified in different ways. For example, LLaMA-2-7B is referred to as "LLaMA-2 7B" and "LLaMA 2 (7 billion)" among other variants. We mapped different ways of referring to the same model to a canonical name.

## 3.2 Benchmarks

We catalog the evaluation methods used in the reviewed research works, focusing on the specific benchmarks, data types, and metrics employed. There were a total of 7 635 different benchmarks used across the 4 886 papers [2]. The total number of benchmark mentions across all papers was 14 970, so on average each benchmark is used in fewer than two papers. The average number of benchmarks used per paper is 3.07.

---

[2]"PromptCARE: Prompt Copyright Protection by Watermark Injection and Verification" (Yao et al., 2023b) mentions evaluation on "LLaMA-3b, LLaMA-7b, and LLaMA-13b" models. There is no LLaMA-3B model released by Meta AI. The original LLaMA release (February 2023) included only 7B, 13B, 30B, and 65B parameter variants, making the reference to a 3B model factually incorrect.

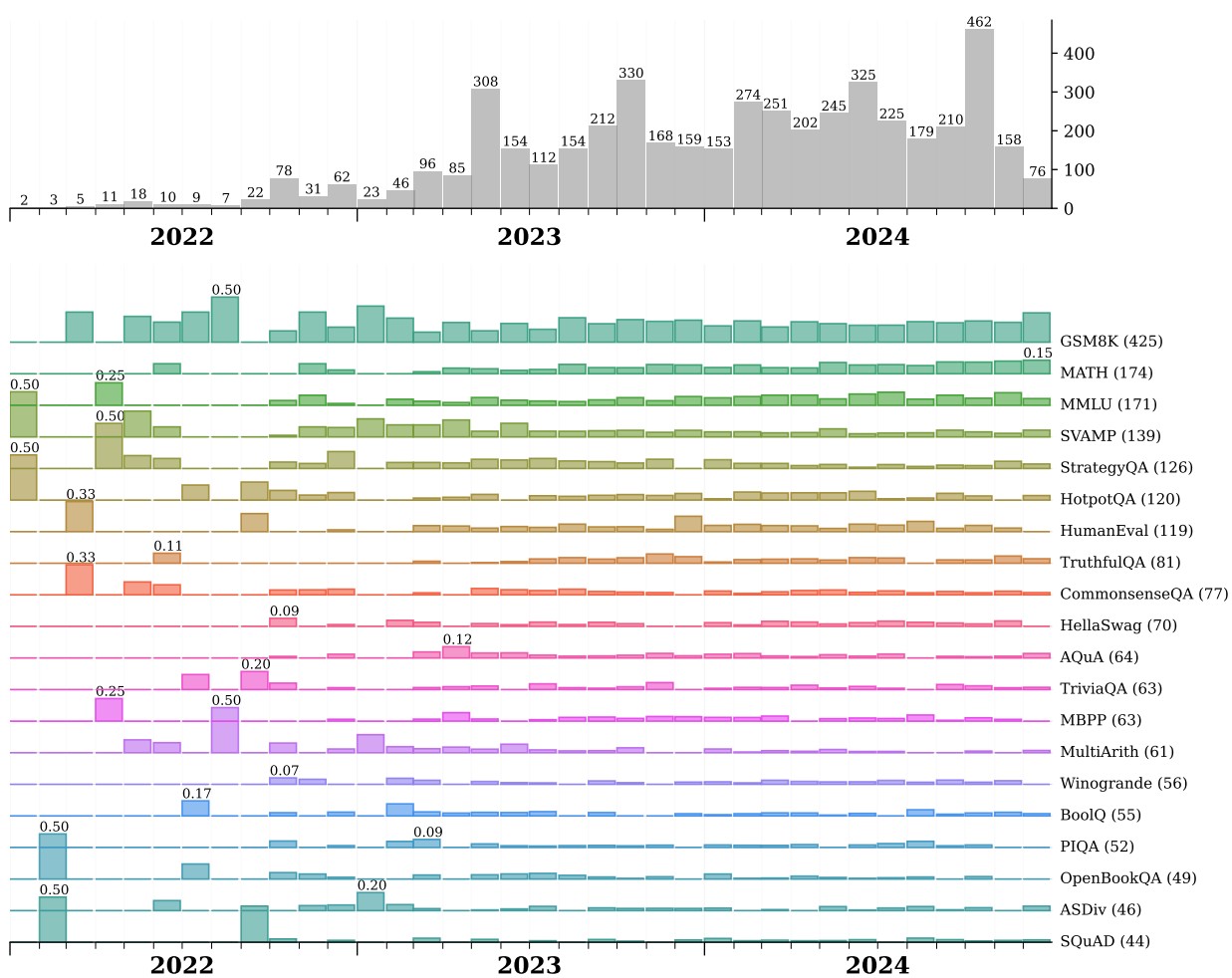

Figure 1: Proportions of papers posted in each month using each of the Top-20 LLM benchmarks (1 Jan 2022–31 Dec 2024). The top plot shows the total number of papers published per month. The bottom plot shows the proportion of the top-20 benchmarks. Each bar represents the proportion of papers in that month that used the given benchmark, calculated as (number of papers using benchmark) / (total benchmark mentions in papers that month). Total uses of each benchmark across the dataset are shown in parentheses.

Table 1 lists the most popular benchmarks used for evaluation across the set of paper. A few popular benchmarks are used in hundreds of papers, but no benchmark is common to more than 10% of the papers. Among the benchmarks, GSM8K (Grade School Math 8K) (Cobbe et al., 2021) is the most widely used, with 425 uses across the papers considered. GSM8K is a dataset of multiple-choice word problems, where the accuracy of the predicted answers is used as the evaluation metric. Several other popular benchmarks primarily test mathematical reasoning including SVAMP (139 uses), HumanEval (119), MBPP (63), MultiArith (61), and ASDiv (Academia Sinica Diverse MWP Dataset) Miao et al. (2020) with 46 mentions.

The most popular benchmarks that do not focus just on mathematical reasoning are MMLU (171 mentions) and StrategyQA (126). MMLU (Multi-task Language Understanding) Hendrycks et al. (2021a) is a large-scale multi-subject benchmark that covers a wide range of academic subjects. StrategyQA (Geva et al., 2021) consists of multiple-choice questions that require reasoning over both a question and a given context to arrive at the correct answer. Other notable benchmarks include HotpotQA (120 mentions), TruthfulQA (81 mentions), and CommonsenseQA (77 mentions). HotPotQA is a dataset of multi-hop question-answering problems that require reasoning over multiple paragraphs. TruthfulQA is a benchmark for evaluating the truthfulness of generated answers, where models are assessed based on their ability to generate truthful

and informative responses. SVAMP (Patel et al., 2021) is another multiple choice dataset of verb argument structure alternations, and the accuracy of predicted verb forms serves as the evaluation method. Figure 1 shows these distribution changes, as different benchmarks become more popular in the research space.

Our analysis reveals a lack of consensus on what benchmarks should be used to evaluate methods for improving LLM reliability. For our experiments in Section 4, we select a representative set of benchmarks to evaluate methods across a diverse set of tasks and domains.

### 3.3 Models

Although there is somewhat more consensus on the models to use for evaluations than there is on the benchmarks, there is still a large variation in the models used with a total of 4 809 distinct models identified across 16 647 total model mentions in the analyzed papers. Although we attempted to canonicalize model names, at least for the commonly used models, the large number of models may be partly due to variations in how the same model is named or very minor variations of a common model.

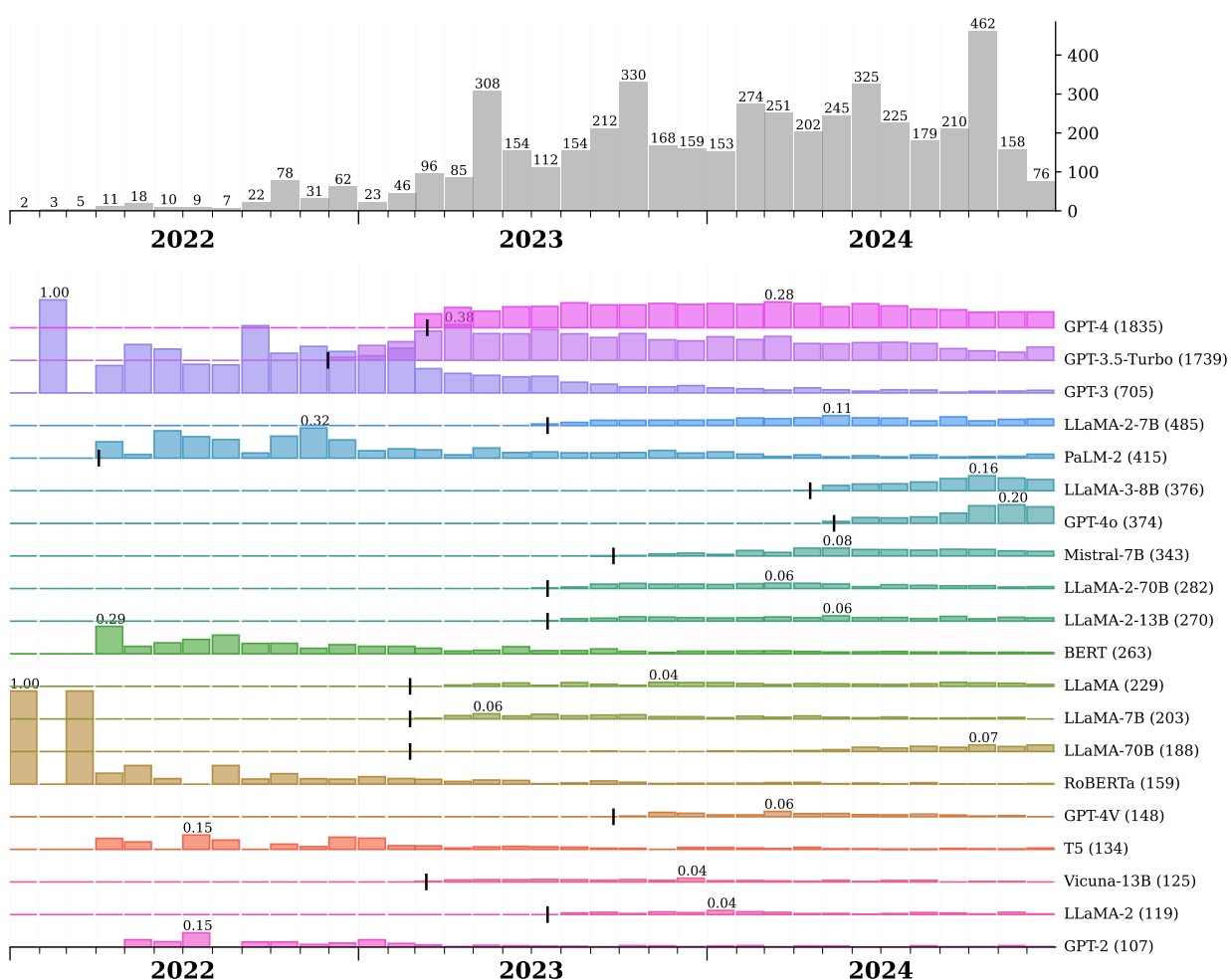

Figure 2: Monthly proportions of papers using each of the Top-20 LLMs (Jan 2022 - Dec 2024). The top plot shows the total number of papers published per month. Each bar in the bottom plot shows the fraction of papers published in that month that used the given model, computed as (papers mentioning model) / (total models that month). Total usage counts across the entire period are shown in parentheses. Each black tick mark demarcates the respective model release date. For consistency across model families, we consolidated model variants (base, chat, and instruct) within each family and size.

The right side of Table 1 summarizes the models most commonly used across the considered papers. The average number of models used per paper is 3.42. GPT-4 is the most frequently mentioned model, appearing in 1835 of the 4886 papers, followed by GPT-3.5-Turbo with 1739 mentions and GPT-3 with 705 mentions. The diversity in model selection even among the top twenty spans from smaller models like T5 (134 mentions) to large open models like LLaMA-2-70B (282 mentions), reflecting the research community's interest in understanding performance across different model scales and architectures. In some cases, the specific size version of the model is not determined by our analysis—there are 119 mentions of LLaMA-2 without specific distinction of the size of the model used, and 229 LLaMA without a version number of size identified, and it is likely that some of the later ones are for LLaMA-2 or LLaMA-3.

Open-weights models feature prominently in the evaluations, with Mistral-7B (343 mentions), LLaMA-2-7B (485 mentions), and various LLaMA variants collectively used in a significant portion of the evaluations. The frequent use of open-weights models enables reproducibility and transparency in a way that is not possible with models only available through an API. Open models also have important cost advantages over models that can only be access through pay-per-use APIs, although the proprietary OpenAI models remain the most popular.

Model selection also changes over time, as new models become available, as shown in Figure 2. While newer models like GPT-4 dominate evaluations through the end of 2024, established models like BERT (263 mentions) and RoBERTa (159 mentions) continue to serve as important baselines. Notably, the emergence of multi-modal models is evident with GPT-4V receiving 148 mentions, highlighting the growing interest in models that can handle both text and visual inputs.

Our analysis highlights key gaps that shape our experimental design. The fragmentation of benchmark usage motivates evaluation across diverse tasks to better assess generalizability. Variability in model choice suggests that reported gains may hinge on specific model traits. Lastly, the underuse of strong, modern models in prior work motivates our focus on evaluating methods with contemporary state-of-the-art systems.

## 4 Experiments

Our review of the research literature revealed variation in the benchmarks and underlying models used to evaluate inference-time methods for improving LLM reliability, motivating us to conduct experiments to assess the robustness and generalizability of proposed methods. Our primary goal is to determine whether results obtained from specific benchmarks and models in previous evaluations hold up when tested more comprehensively across a diverse range of state-of-the-art language models and previously unseen test data.

In our experiments, we evaluate the performance of the selected methods on different LLMs including more recent and powerful language models than may have been used in the original evaluations. We also test proposed methods on a broader set of benchmarks, including both commonly and rarely used benchmarks. Our experiments also assess the consistency of inference-time method performance across different model architectures and types. With these experiments, we aim to provide a more nuanced understanding of the effectiveness of current techniques, identifying both strengths and limitations that may not have been apparent in original evaluations.

### 4.1 Methods Evaluated

Given our limited resources, we were not able to include all methods in our experiment. We selected methods to evaluate from papers in the literature review based on availability of standard implementations with a goal of having a representative set of methods to test. We conduct experiments using the following methods, ordered by the date that the methods were first posted on arXiv: Chain of Thought (Wei et al., 2022), Self-Consistency Wang et al. (2023c), ReAct (Yao et al., 2023d), Tree of Thoughts (Yao et al., 2024), Graph of Thoughts (Besta et al., 2024), and LLM Multi-Agent Debate (Du et al., 2024). We provide descriptions of each of these methods and how we configured them for our experiments below.

The selected methods range from the basic prompt augmentation and output aggregation in Chain of Thought to complex multi-model interactions and sophisticated aggregation in LLM Multi-Agent Debate. Although

Chain of Thought involves manually designing prompts to guide the model through step-by-step reasoning, we include the Chain of Thought method as a baseline to compare its effectiveness against the other automated inference-time techniques. We emphasize that our goal is not to identify the "best" inference-time method, hence this limited but representative selection of methods, but rather to understand what is necessary to perform a robust evaluation of an inference-time method.

For each method, we used the default settings provided by their respective repositories to ensure reproducibility and maintain consistency with the original implementations. This means the same language models may have different hyper-parameters when used across different methods. While this could potentially introduce confounding factors in performance trends, we prioritize fidelity to the original implementations as reported in their respective papers and repositories to allow for a more direct comparison with previously published results. Full prompt structures for each method are found in Appendix C.

In evaluating some of the methods, reproducing and running results proved challenging due to resource limitations, compatibility issues with current AI models, and the use of outdated or unsupported models in original evaluations. Some methods required excessive processing time or computing power, while others used packages incompatible with state-of-the-art models like those from Anthropic and newer OpenAI versions. Some methods include Pitis et al. (2023), Arora et al. (2022), and Si et al. (2023). We take these into consideration when comparing methods, models, and benchmarks and providing recommendations.

**Chain of Thought** (Wei et al., 2022). Chain of Thought is a method designed to enhance the reasoning abilities of large language models. Each exemplar in few-shot prompting is augmented with a series of intermediate natural language reasoning steps—that leads to the final answer. The method samples from the output using greedy decoding. The original experiments used multiple arithmetic reasoning benchmarks (GSM8K, SVAMP, ASDiv, MAWPS, and AQuA), and several models reflecting the state-of-the-art at the time: GPT-3 (350M–175B parameters), LaMDA (422M–137B), PaLM (8B–540B), UL2 (20B), and Codex.

For evaluating models on multiple-choice benchmarks, we used a Chain of Thought implementation based on the approach outlined in the referenced repository (Yao et al., 2023a). For more generative benchmarks, we used the method outlined in Besta et al. (2024). Each of the models were configured with a temperature of 0.7 and a maximum token limit of 1024 to allow for more elaborate reasoning chains. The prompt included multiple stages, with the model first analyzing the problem, laying out intermediate thought processes, and then computing or inferring the final result. We use "Let's think step by step" as a leading instruction guided the model in decomposing tasks into manageable chunks. More information on the prompt structure is found in Appendix C.

**Self-Consistency** (Wang et al., 2023c). Self-consistency is an enhancement to Chain of Thought prompting that aims to improve language models' performance on complex reasoning tasks. While standard Chain of Thought uses greedy decoding to generate a single reasoning path, self-consistency samples multiple diverse reasoning paths from the model. It then extracts the final answer from each path and determines the most consistent answer through majority voting. The authors evaluated self-consistency on a variety of arithmetic and commonsense reasoning benchmarks (GSM8K, SVAMP, AQuA, StrategyQA, and ARC-challenge) with four underlying LLMs—PaLM (540B), GPT-3 (175B), LaMDA (137B), and UL2 (20B).

The original authors did not provide a public implementation, so we produced our own implementation following the description in the paper to include this in our experiments. The method was applied by generating 3, 5, and 10 diverse reasoning paths for each task. These reasoning paths were produced by prompting the model multiple times, with a focus on encouraging variation in the intermediate steps taken toward the solution. The prompt augmentation phase utilizes Chain of Thought prompt methods. Each path was evaluated independently, and the final answer was determined by aggregating the results to select the most consistent outcome across all generated paths. The process allowed the model to explore a range of potential solutions, increasing the likelihood of arriving at a good answer through collective reasoning.

**ReAct** (Yao et al., 2023d). The ReAct method combines reasoning and acting by augmenting a language model's action space to include both external actions and language-based thoughts. For each seed prompt, the model generates a series of thoughts that update the context without affecting the environment. The

model's responses are based on few-shot in-context examples, each containing a human-generated trajectory of actions, thoughts, and observations for a specific task instance. The original study tested the method using PaLM-540B and GPT-3 on the HotpotQA, Fever, Alfworld, and WebShop benchmarks. Subsequent ablation studies also included testing GPT-3.5-turbo on the GSM8K dataset (Face, 2023).

For our experiments, ReAct was implemented using a custom LangChain agent based on the original reference repository Chase (2022). Format errors were corrected during execution to ensure the output followed the required structure. The model was configured with a temperature of 0.5, a maximum token limit of 512, and up to two retries in case of errors. It was instructed to stop generating text at specific markers such as "\nHuman:" or "Final Answer:".

**Tree of Thoughts** (Yao et al., 2024). While Chain of Thought generates a single sequence of thoughts and lacks exploration of alternative reasoning paths and Self-Consistency improves upon this by sampling multiple independent chains, Tree of Thoughts frames problem-solving as a search over a tree of thoughts, allowing for both local and global exploration of the problem space. This paradigm incorporates planning, look-ahead, and backtracking, enabling the evaluation and pruning of intermediate states. The Tree of Thought framework consists of four key components: thought decomposition, thought generation, state evaluation, and search algorithms. Thought decomposition breaks down the intermediate process into discrete steps, adapting to different problem types. The thought generator produces $k$ candidates for the next thought given a tree state. It employs two strategies: sampling, which uses a Chain of Thought prompt to generate thoughts, and proposing, which uses a "propose prompt" to sequentially generate thoughts. The state evaluator then assesses the progress of different states towards solving the problem, serving as a heuristic for the search algorithm. It can either value each state independently or vote across states, depending on the problem's characteristics. The search algorithm navigates the tree structure to find the solution, using either a breadth-first search (BFS) or depth-first search (DFS) strategy. BFS maintains a set of the most promising states per step and is used for problems with limited tree depth, while DFS explores the most promising state first until a final output is reached or deemed impossible to solve.

For benchmarks based on multiple-choice question answering solutions, we used a Tree of Thought implementation based on the approach outlined in the referenced repository (Yao et al., 2023c). For more generative benchmarks, we used the method outlined in Besta et al. (2024). To run the experiments, we used the standard prompt format to generate sequences of thoughts through sequential proposals, where each intermediate thought was sampled based on previous reasoning steps, creating a more structured exploration. For this, the model used a propose strategy for thought generation and a value approach for evaluating the intermediate states. During evaluation, each path was sampled multiple times, with the system generating thoughts multiple times to improve diversity. The default parameters prompted the model to generate sequences 10 times and evaluate each generated state five times, while the BFS algorithm was configured to retain the top three states at each step to explore further.

**Graph of Thoughts** (Besta et al., 2024). Graph of Thoughts introduces a graph-based structure to model reasoning paths, allowing for more flexible exploration of possible solutions by dynamically connecting paths. Graph of Thoughts models the LLM's reasoning process as an arbitrary graph, where thoughts are represented as vertices and dependencies between thoughts as edges. This graph-based approach allows for more complex transformations than previous methods like Chain of Thought or Tree of Thoughts. The framework is modeled as a tuple containing the reasoning process graph, "thought" transformations, an evaluator function, and a ranking function. Graph of Thoughts enables several graph-enabled transformations, including thought aggregation, where multiple thoughts can be combined into new ones, thought refinement through iterative improvement, and parallel thought generation. These transformations are managed through a scoring and ranking system that evaluates thoughts and selects the most promising forward paths. For our experiments, we used the open-source repository (Blach et al., 2023).

**LLM Multi-Agent Debate** (Du et al., 2024). In an LLM Multi-Agent Debate, multiple agents are prompted to evaluate a problem from different perspectives, and through a series of interactions, they converge on a solution through collective reasoning. For our experiments we used the repositories (Du et al., 2023) for proprietary models and (Gauss5930, 2023) for open-source models. Furthermore, we utilized three

agents, all based on the same model, and conducted two rounds of debate per task. During each round, the agents would present their arguments, and subsequent rounds allowed them to refine or rebut each other's points. The debate was structured to ensure that the agents were working both collaboratively and competitively to arrive at the most accurate solution. The response was determined based on the consensus or the strongest argument presented by the agents at the end of the debate. While the same model was used for all agents in this study, future research could explore the potential benefits of using a diverse range of models as agents, allowing for even greater variation in reasoning and argumentation.

## 4.2 Setup

For our experiments, we selected a set of representative methods to evaluate and a common set of models and benchmarks to use in the evaluation, informed by our literature analysis from Section 3.

**Models.** As reported in Table 1, many different models have been used in evaluations, and most works only evaluate on three different models, averaged across papers. For our experiments, we selected a mix of models to enable us to measure both the performance of each method on state-of-the-art models and how that performance varies with smaller models. For the state-of-the-art models, we selected GPT-4o and Claude 3.5-Sonnet. As of May 2024 (when we started our experiments) the two models performed well in the LMSYS Chatbot Arena Leaderboard Chiang et al. (2024). Although more recent models have now surpassed these models in most rankings, both GPT-4o and Claude 3.5-Sonnet are still reasonably highly-ranked in LMSYS and other rankings such as the SCALE AI rankings (Scale AI, 2025). We include GPT-3.5-turbo to compare our results with most of the reported results across methods. We include two open-weights models, Llama-3.1-8B-Instruct and Mixtral 8x22B, both for comparison and to enable reproducibility.

**Benchmarks.** For our analysis, we selected five of the most commonly used benchmarks in the literature—GSM8K, MMLU, AQuA, SVAMP, TruthfulQA. These benchmarks are described in Section 3.2, and include three popular mathematical reasoning benchmarks (GSM8K, AQuA, and SVAMP) and two broad language understanding benchmarks (MMLU and TruthfulQA).

We also include GSM-Symbolic (Mirzadeh et al., 2024), a relatively new benchmark that was not used by any of the papers in our analysis. GSM-Symbolic was constructed by converting GSM8K questions into symbolic templates that allow for controlled variation of parameters like names, numbers, and problem complexity. Using 100 templates from GSM8K, it generates 50 samples per template, resulting in 5 000 total examples for each benchmark variant. The dataset includes different difficulty levels, from simpler versions with clauses removed (GSM-Symbolic-M1) to more complex versions with additional clauses (GSM-Symbolic-P1, P2), and a special variant (GSM-NoOp) that tests models' ability to identify relevant information. The dataset enables evaluation of models' robustness to parameter changes, handling of increasing complexity, and true understanding of mathematical concepts versus pattern matching. Our evaluation follows the methodology from Mirzadeh et al. (2024).

We include two domain-focused benchmarks, MedQA (Jin et al., 2021) and LegalBench (Guha et al., 2023; Koreeda & Manning, 2021; Hendrycks et al., 2021b; Wang et al., 2023b; Wilson et al., 2016; Zheng et al., 2021; Zimmeck et al., 2019; Ravichander et al., 2019; Holzenberger & Van Durme, 2021; Lippi et al., 2019). We sample from the MedQA dataset, which contains multiple-choice questions from medical board exams across three languages. For LegalBench, we only use the "privacy_policy_qa" subset, which focuses on question answering related to privacy policies, drawn from a larger benchmark of 162 legal reasoning tasks.

We include two additional benchmarks that were not commonly used, but were chosen to further evaluate generalization. Sorting 032 (Besta et al., 2024) evaluates a model's ability to sort a sequence of numbers in ascending order. It consists of array sequences of 32 numbers in the range 0–9, and measures the model's performance as the percentage of correctly sorted sequences. Document Merging was introduced in the Graph of Thoughts paper (Besta et al., 2024). It assesses a model's ability to merge multiple documents into a single coherent document. To evaluate an output, we use the same evaluation criteria as in Besta et al. (2024). We use the underlying LLM in each experiment to assess two key metrics, each queried three times to obtain an average. The first metric measures conciseness, with 0 suggesting at least 50% redundant information and 10 indicating no redundancy. The second metric gauges information preservation, where 0

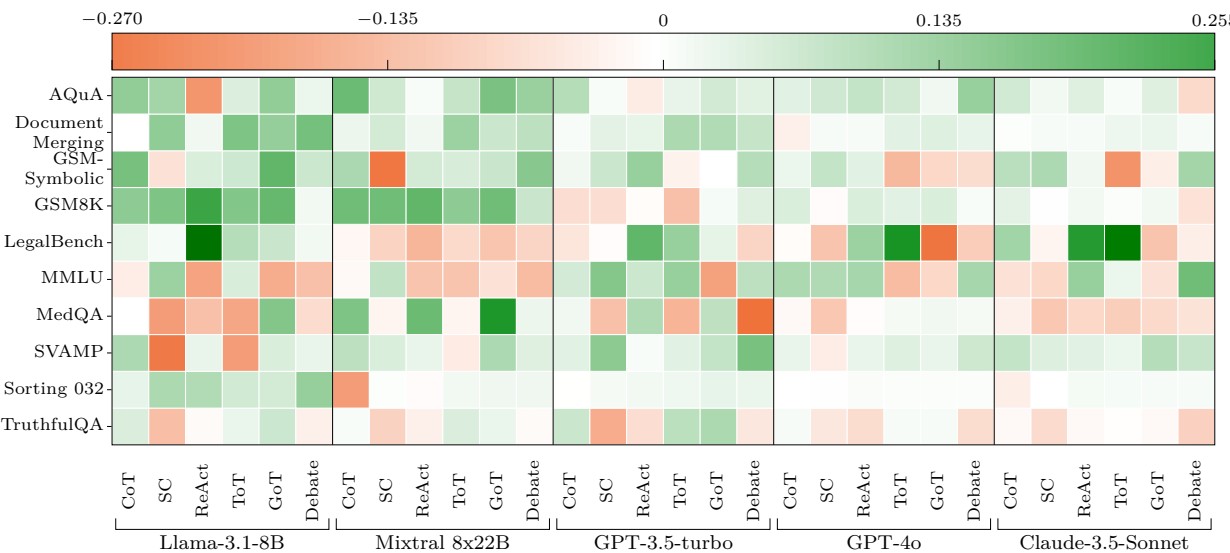

Figure 3: Heatmap displaying the deviation from baseline accuracy for various inference-time methods applied across different models and benchmarks. Positive deviations (in green) indicate improvements over the unaided model (baseline), while negative deviations (in red) indicate performance decline.

indicated total information loss and 10 signifies complete retention. The final score is the harmonic mean of these two values and the average rating across multiple document sets is used as the evaluation metric.

# 5 Results

A key goal of our experiments is to understand how well results from reported benchmarks and models predict the performance of a method on other benchmarks and with better models. Section 5.1 reports on the overall performance of the inference-time methods. Section 5.2 considers how well results in previous evaluations are reproduced in our experiments, and Section 5.3 evaluates the how the execution costs vary.

## 5.1 Performance of Inference-Time Methods

Figure 3 compares the results from different methods using different models on the selected benchmarks and Table 2 summarizes the average performance improvement for each method across the benchmarks across the five evaluation models.

Due to the high cost of running some of the methods (which we discuss in Section 5.3), for each of the methods we randomly sample 150 data points from each of benchmarks. We then normalize the data by employing a min-max normalization to standardize data across different methods, scaling all values to a 0–1 range to support consistent comparisons across benchmarks. The *baseline* refers to the accuracy achieved by each model when no inference-time method is employed. Positive deviations, colored green in the heat map, indicate improvements over the unaided model baseline, whereas negative deviations, shown in red, indicate declines in performance.

We observe that significant variations in performance across different tasks, models, methods, and benchmarks, which remains a problem in evaluating large language models and respective ensemble methods. Note in particular that each method has a *negative* impact on at least one of the benchmarks. Two of the benchmarks (GSM8K, Document Merging) exhibit positive improvements on average for all of them methods, but for every other benchmark at least one of the methods results in a performance reduction for at least one of the base models.

Table 2: Average performance change across all models and methods. Positive values indicate improved performance relative to a baseline. Results show varying effectiveness of each technique, highlighting the challenge of developing methods that work across different problem types. Cost is measured in average number of API calls per problem for GPT-3.5-turbo.

| | CoT | SC | ReAct | ToT | GoT | Debate |
|---|---|---|---|---|---|---|
| Relative Cost | 3.5 | 10.0 | 5.2 | 53.8 | 37.2 | 6.0 |
| AQuA | $0.110 \pm 0.057$ | $0.057 \pm 0.040$ | $-0.024 \pm 0.102$ | $0.045 \pm 0.024$ | $0.090 \pm 0.062$ | $0.054 \pm 0.081$ |
| Document Merg | $0.001 \pm 0.019$ | $0.054 \pm 0.051$ | $0.019 \pm 0.008$ | $0.096 \pm 0.057$ | $0.079 \pm 0.042$ | $0.079 \pm 0.060$ |
| GSM-Symbolic | $0.087 \pm 0.060$ | $-0.015 \pm 0.143$ | $0.062 \pm 0.041$ | $-0.053 \pm 0.112$ | $0.034 \pm 0.102$ | $0.077 \pm 0.079$ |
| GSM8K | $0.073 \pm 0.092$ | $0.059 \pm 0.105$ | $0.107 \pm 0.110$ | $0.048 \pm 0.107$ | $0.096 \pm 0.086$ | $0.017 \pm 0.044$ |
| LegalBench | $0.017 \pm 0.062$ | $-0.045 \pm 0.051$ | $0.181 \pm 0.190$ | $0.171 \pm 0.162$ | $-0.082 \pm 0.126$ | $-0.058 \pm 0.044$ |
| MMLU | $-0.024 \pm 0.120$ | $0.099 \pm 0.054$ | $-0.031 \pm 0.114$ | $-0.002 \pm 0.099$ | $-0.147 \pm 0.068$ | $-0.008 \pm 0.107$ |
| MedQA | $0.030 \pm 0.074$ | $-0.115 \pm 0.057$ | $0.019 \pm 0.120$ | $-0.088 \pm 0.075$ | $0.101 \pm 0.129$ | $-0.075 \pm 0.112$ |
| SVAMP | $0.070 \pm 0.030$ | $-0.011 \pm 0.143$ | $0.028 \pm 0.010$ | $-0.026 \pm 0.092$ | $0.074 \pm 0.030$ | $0.080 \pm 0.054$ |
| Sorting | $-0.041 \pm 0.082$ | $0.026 \pm 0.044$ | $0.028 \pm 0.040$ | $0.024 \pm 0.020$ | $0.026 \pm 0.019$ | $0.042 \pm 0.051$ |
| TruthfulQA | $0.026 \pm 0.031$ | $-0.102 \pm 0.042$ | $-0.038 \pm 0.026$ | $0.036 \pm 0.033$ | $0.042 \pm 0.044$ | $-0.050 \pm 0.029$ |

For the Llama-3.1-8B-Instruct model, we observe significant improvements (visible in Figure 3) across most benchmarks when using inference-time methods. The Graph of Thoughts method shows particularly strong performance on the Document Merging task (which was introduced in the Graph of Thoughts paper). Chain of Thought and Tree of Thoughts also demonstrate consistent improvements across various benchmarks for this model. Mixtral 8x22B shows a more varied performance profile. While it benefits from inference-time methods in tasks like AQuA and GSM8K, it shows some negative deviations in benchmarks such as MMLU and TruthfulQA. The Multi-Agent Debate approach appears particularly effective for this model on the AQuA benchmark. GPT-3.5-turbo demonstrates more modest improvements from ensemble methods compared to the previous two models. However, it still shows positive deviations in several benchmarks, particularly when using the Self-Consistency method on GSM8K and the ReAct method on AQuA.

For the more advanced models, GPT-4o and Claude-3.5-Sonnet, the impact of the tested methods is less pronounced, as visible in the lighter color shades in the heat map. This suggests that these models already perform well on many tasks without additional ensemble techniques and obtaining further improvements through inference-time methods is more challenging. We do observe some improvements, particularly in the Document Merging task for both models when using the Graph of Thoughts method (which introduced this benchmark). The SVAMP benchmark shows interesting variations across models and methods. While some ensemble methods yield improvements for Llama-3.1-8B and Mixtral 8x22B, the more advanced models show minimal changes or even slight negative deviations when applying these methods to SVAMP.

Overall, our experiments reveal that the effectiveness of tested methods varies not only across different benchmarks but also across different model capacities. Less powerful models like Llama-3.1-8B-Instruct tend to benefit more consistently from the inference-time methods, while more advanced models like GPT-4o and Claude-3.5-Sonnet show limited improvements.

## 5.2 Reproducibility

In addition to understanding how well methods generalize to different models and benchmarks, we also wanted to study now reliably results reports in papers could be reproduced.

We compare GPT-3.5-turbo's performance both without any inference-time method (Unaided) to the tested method (Using Method) in Table 3, showing both the results reported in the original papers and our reproductions and generalizations. We chose GPT-3.5-turbo for its availability, reproducibility, and suitability for cost estimation, and validated that relative cost patterns across methods remain consistent across models.

Table 3: Comparison of reported and reproduced results across different methods and benchmarks. The Chain of Thought reported results are from the "Chain-of-Thought Hub" repository (Fu et al., 2023). The Tree of Thoughts (Yao et al., 2024) GSM8K reported results are from the original GPT-4 experiments. The Self-Consistency reported results are from experiments with GPT-3 (Wang et al., 2023c). All reproduced results are from our own experiments using GPT-3.5-turbo.

| | GPT-3.5-turbo | | | |
| | Unaided | | Using Method | |
| Methods and Benchmarks | Reported | Reproduced | Reported | Reproduced |
|---|---|---|---|---|
| **Chain of Thought** | | | | |
| GSM8K | – | 0.63 | 0.75 (+0.12) | 0.70 (+0.07) |
| MMLU | – | 0.63 | 0.67 (+0.04) | 0.69 (+0.06) |
| AQuA | – | 0.64 | – | 0.74 (+0.10) |
| SVAMP | – | 0.72 | – | 0.76 (+0.04) |
| Sorting 032 | 0.86 | 0.96 (+0.10) | 0.79 (-0.17) | 0.96 (+0.00) |
| Document Merging | 0.64 | 0.70 (+0.06) | 0.66 (-0.04) | 0.71 (+0.01) |
| **Self-Consistency** | | | | |
| GSM8K | – | 0.63 | – | 0.70 (+0.07) |
| MMLU | – | 0.63 | – | 0.79 (+0.16) |
| AQuA | – | 0.63 | – | |
| SVAMP | – | 0.72 | – | 0.87 (+0.15) |
| **ReAct** | | | | |
| GSM8K | – | 0.63 | – | 0.76 (+0.13) |
| **Tree of Thoughts** | | | | |
| GSM8K | – | 0.63 | – | 0.64 (+0.01) |
| Sorting 032 | 0.86 | 0.96 (+0.10) | 0.95 (-0.01) | 0.98 (+0.02) |
| Document Merging | 0.64 | 0.70 (+0.06) | 0.78 (+0.08) | 0.81 (+0.11) |
| **Multi-Agent Debate** | | | | |
| GSM8K | 0.77 | 0.63 (-0.14) | 0.85 (+0.22) | 0.81 (+0.18) |
| MMLU | 0.64 | 0.63 (-0.01) | 0.71 (+0.08) | 0.72 (+0.09) |

Appendix B provides more discussion on model choice. Table 3 reveals discrepancies between the reported and reproduced results for GPT-3.5-turbo, underscoring reproducibility challenges in evaluating large language models.

**Detailed view of Document Merging benchmark.** Figure 4 shows a more detailed view of the impact of the methods on the Document Merging benchmark. This task serves as a useful benchmark to compare the performance of different language models as it represents a common use case across various LLM applications. While this benchmark was not originally used in the papers introducing Chain of Thought and Tree of Thoughts, the authors of the Graph of Thoughts paper instantiated their own replications of these methods for comparison purposes. This explains the discrepancy between the benchmark data presented in Table 3. The Document Merging benchmark thus offers a unique perspective on how these different approaches perform when implemented under consistent conditions by the same research team.

We first replicate the results reported by Besta et al. (2024) for this task using GPT-3.5-turbo. We then extend our evaluation to use our selected models. Our analysis reveals interesting patterns in model performance. While state-of-the-art models like GPT-4o and Claude 3.5-Sonnet show negligible improvements over the baseline, the magnitude of these gains varies. These results not only demonstrate the varying capabilities of different models in handling complex tasks like document merging but also highlight the importance of model selection for specific applications. The findings suggest that while more powerful models generally

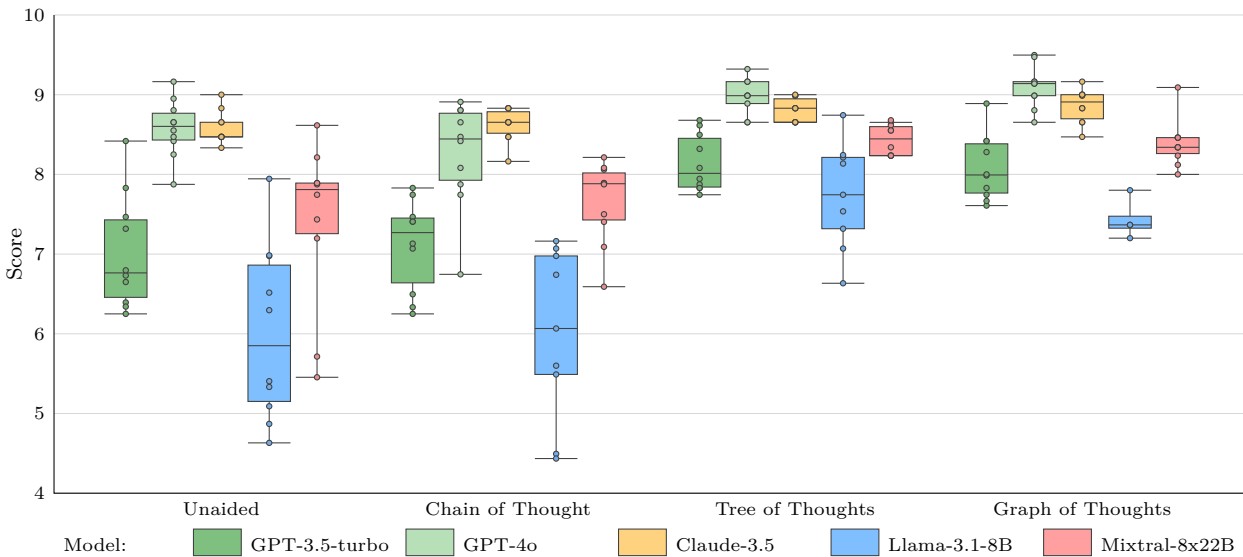

Figure 4: Comparison of document merging task across different models and approaches Besta et al. (2024). The plot illustrates the performance of different approaches (Unaided, Chain of Thought, Tree of Thoughts, and Graph of Thoughts), using box plots to summarize score distributions and swarm plots to show scores of individual iterations.

perform better, the degree of improvement can vary based on the task complexity and the specific strengths of each model.

### 5.3 Cost

Although our analysis focused on reliability, the cost of executing an inference-time method can be substantial. We measure costs by the number of API calls used to solve a given problem in a benchmark and include these results in Table 2.

It is clear that the cost of all of the methods are high, requiring an average of from 3.5 (Chain of Thought) to over 50 (Tree of Thoughts) instantiations of the base LLM for each seed prompt. We use API calls to measure cost due to the changing nature of token costs for black box models. We report the number of calls when the base model is GPT-3.5-turbo, and the multiple varries only slightly based on the base model used (see Appendix B for analysis supporting this claim).

Some recent works have explored ways to reduce these costs. Snell et al. (2024) studies how to optimally scale test-time computation for LLMs on math reasoning tasks, while Chen et al. (2024) examines the cost-benefit tradeoffs of multiple API calls. These findings align with our recommendations about the importance of balancing performance gains against computational costs.

As mentioned earlier, these high costs limited our experiments, but they are an important factor in any considered deployment. One measure of the practical value of these methods would be if the total cost of obtaining the same performance is lower using the inference-time method with a less expensive model than the cost of obtaining similar performance from a state-of-the-art high cost model.

## 6 Discussion

Improving LLM reliability is a critical goal, and there is an active research community exploring myriad approaches, including much focus on the inference-time methods we study here. To make progress in this area, it is critical that evaluations are done in a way that can robustly determine if a proposed method is a meaningful improvement on other methods.

Our analysis of the evaluation approaches used in the considered literature shows a large range of different evaluation methods, with hundreds of different benchmarks used and more than half of the papers conducting evaluations with just one benchmark and no benchmark used by more than a quarter of the evaluations (although these are primarily the result of surveys in the citation chain). There is somewhat more agreement on the models to use, and the available state-of-the-art proprietary and open weights models will continue to change over time.

As demonstrated in our experiments, and captured in Figure 3, the impact of an inference-time method on reliability varies substantially across both underlying models and selected benchmarks. Methods that produce large improvements with weaker models often produce little improvement (or even make things worse) for stronger models. As further emphasized by Table 2, methods that produce significant improvements for certain mathematical reasoning benchmarks, may not result in improvements for other benchmarks.

Our results highlight the importance of evaluating methods with a range of underlying models, but especially with models representative of the state-of-the-art, at least if the goal is to develop methods that are useful in making overall improvements in reliability. The choice of benchmarks is also important. Unfortunately, the resources required to run extensive tests on all available benchmarks are not available to researchers outside of the largest industry groups, so it is important to select a suite of benchmarks that are sufficient to understand the impact of a method across a range of settings that cover the intended use cases. None of the current benchmarks, at least of the ones considered in our evaluation, are representative enough to be used as a single benchmark that would allow researchers to draw general conclusions.

Based on our findings, we propose several recommendations to address the variability in inference-time method performance across models and benchmarks.

First, researchers should evaluate across diverse models. Methods that demonstrate improvements on weaker models often underperform or even degrade results when applied to stronger models. Researchers should assess their methods across a range of model capabilities, including state-of-the-art systems, rather than relying solely on older or less capable baselines.

Second, researchers should select a representative suite of benchmarks. No single benchmark reliably captures the performance of an inference-time method across different domains and reasoning types. Researchers should curate diverse benchmark collections that span various task formats and reasoning requirements.

Third, researchers should prioritize benchmark diversity over quantity. Evaluating methods on a narrow set of similar benchmarks risks overfitting to specific task characteristics. A strategic selection provides more balanced assessments of method effectiveness than exhaustive testing on numerous similar tasks.

Finally, researchers should adopt standardized evaluation protocols, The wide variation in evaluation approaches undermines comparability across studies. We recommend that researchers minimally include results from widely-used benchmarks alongside additional domain-specific evaluations, supported by clear justifications for benchmark selection.

Much work is needed in understanding the effectiveness of different benchmarks and underlying models to predict the performance of a method in other settings. Developing new benchmarks is part of this, but it is important that the predictive value of any new benchmark is evaluated by also using more established benchmarks. Research will accelerate in this area as the community makes progress to a set of standardized benchmarks and better understanding of how performance impacts translate across models and tasks.

## Acknowledgments

This work is supported in part by funds provided by the National Science Foundation, Department of Homeland Security, and IBM through the ACTION AI Institute (Award #2229876). Any opinions, findings, and conclusions or recommendations expressed in this material are those of the authors and do not necessarily reflect the views of the National Science Foundation or its federal agency and industry partners. The authors would also like to thank Anshuman Suri for his helpful comments and suggestions.

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

## A    Citation Overlap Analysis of Foundational Papers

To validate our choice of Wei et al. (2022) as the primary citation graph, we analyzed the citation overlap between this foundational work and three other influential papers in the field: Retrieval-Augmented Generation for Knowledge-Intensive NLP Tasks (Lewis et al., 2020), Self-Instruct: Aligning Language Models with Self-Generated Instructions (Wang et al., 2023d), and Solving Quantitative Reasoning Problems with Language Models (Lewkowycz et al., 2022). Figure 5 presents a comprehensive visualization of the citation relationships between these works.

The analysis reveals several key insights about the citation landscape in this research area. First, the Chain-of-Thought paper demonstrates the broadest citation coverage with 8,887 total citations, significantly exceeding the other works (RAG: 5,970; Self-Instruct: 2,134; Quantitative Reasoning: 783). This substantial citation base provides a comprehensive foundation for our analysis.

The overlap patterns indicate meaningful but distinct research focuses. Chain-of-Thought shares 975 citations with RAG (16.3% of RAG's citations), suggesting complementary approaches to enhancing language model capabilities. The overlap with Self-Instruct (423 shared citations, 19.8% of Self-Instruct's total) reflects shared foundations in instruction-following and reasoning tasks. Most notably, the Quantitative Reasoning paper shows the highest proportional overlap with Chain-of-Thought (355 shared citations, 45.3% of its total), indicating strong alignment in mathematical and logical reasoning literature.

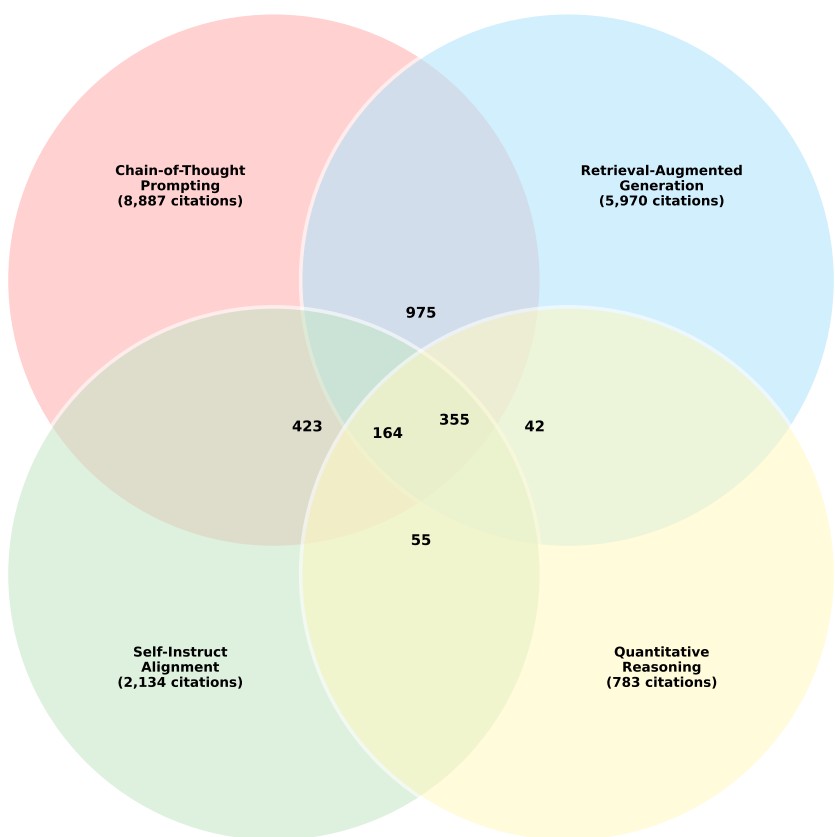

Figure 5: Citation overlap analysis showing shared references between foundational papers in large language model research. Numbers indicate shared citations between paper pairs, with total citation counts shown for each work.

## B    Reproducibility and Cost Estimation

For the cost analysis, our primary interest was understanding how inference-time methods influence computational cost, measured via the number of API calls. Our assumption was that this metric would remain relatively stable across models for a fixed method. To validate this, we conducted a smaller-scale experiment across five models using 450 problems per method, summarized in Table 4. While absolute values vary, the relative cost pattern across methods remains consistent, supporting our use of GPT-3.5-turbo as a representative model.

Table 4: Average number of API calls per method across models (450 problems)

| Model | CoT | SC | ReAct | ToT | GoT | Debate |
|---|---|---|---|---|---|---|
| GPT-3.5-turbo | 3.5 | 10.0 | 5.20 | 53.8 | 37.2 | 6.0 |
| GPT-4o | 3.5 | 10.0 | 2.07 | 53.8 | 37.2 | 6.0 |
| Claude 3.5-Sonnet | 3.5 | 10.0 | 2.50 | 53.8 | 37.2 | 6.0 |
| Llama-3.1-8B-Instruct | 3.5 | 10.0 | 6.76 | 53.8 | 37.2 | 6.0 |
| Mixtral 8x22B | 3.5 | 10.0 | 4.21 | 53.8 | 37.2 | 6.0 |

For the reproducibility analysis, we acknowledge that Section 5.2 considered only GPT-3.5-turbo, unlike other parts of our evaluation where we use several models. Our decision to focus solely on GPT-3.5-turbo for the reproducibility analysis was driven by practical necessity rather than preference. After reviewing the landscape of available models suitable for our reproducibility analysis, GPT-3.5-turbo emerged as effectively the only viable option. Most models used in the original papers are either deprecated, inaccessible, or unstable for long-term research.

The majority of models used in the original papers, such as `text-ada-001`, `text-babbage-001`, `code-davinci-001`, LaMDA, and PaLM have been deprecated or are no longer publicly available. Using GPT-3.5-turbo provided a consistent baseline across all methods, and its use has become standard for implementing these techniques in current research

Our decision was further necessitated by OpenAI's deprecation schedule. As documented by OpenAI, all legacy models including `text-davinci-002` and `text-davinci-003` were officially shut down on January 4, 2024, with GPT-3.5-turbo-instruct designated as the recommended replacement.[3] GPT-3.5-turbo emerged as the most viable option that balances availability, capability, and consistency across methods.

As shown in Table 5, the models used in the original method papers are largely unavailable or deprecated. GPT-3.5-turbo offers sufficient capability, broad adoption, long-term availability, consistent API behavior, and support for all evaluated methods, making it the only viable baseline for reproducibility-oriented analysis.

As this table demonstrates, GPT-3.5-turbo is the only model that offers both the necessary capabilities and long-term availability for reproducible research across all these methods.

Table 5: Models used in original evaluation for each of the evaluated methods

| Method | Models Used |
|---|---|
| Chain of Thought | GPT-3 (`text-ada-001`, `text-babbage-001`, `text-curie-001`, `text-davinci-002`), LaMDA, PaLM, UL2 20B, Codex |
| Self-Consistency | UL2, GPT-3 (`code-davinci-001`, `code-davinci-002`), LaMDA-137B, PaLM-540B |
| ReAct | PaLM-540B, GPT-3 (`text-davinci-002`) |
| Tree of Thought | GPT-4, GPT-3.5 |
| Graph of Thought | GPT-3.5 |
| Debate | GPT-3.5 |

---

[3]https://platform.openai.com/docs/deprecations

## C   Experimental Prompts

---

**ReAct Prompt Example**

You are an AI assistant designed to answer multiple-choice questions. Analyze the problem and select the best answer from the provided options (A, B, C, D).
Question:
Options: A: B: C: D:
Previous steps:
Your task is to provide a final answer.
Use the following format: Thought: [Your reasoning] Final Answer: [Letter choice (A, B, C, or D)] [Brief explanation]
If you cannot determine the answer with certainty, make your best guess based on the information available.
Your response:

---

**Chain of Thought Prompt Example**

Question: Angelo and Melanie want to plan how many hours over the next week they should study together for their test next week. They have 2 chapters of their textbook to study and 4 worksheets to memorize. They figure out that they should dedicate 3 hours to each chapter of their textbook and 1.5 hours for each worksheet. If they plan to study no more than 4 hours each day, how many days should they plan to study total over the next week if they take a 10-minute break every hour, include 3 10-minute snack breaks each day, and 30 minutes for lunch each day? Let's think step by step Angelo and Melanie think they should dedicate 3 hours to each of the 2 chapters, 3 hours x 2 chapters = 6 hours total. For the worksheets they plan to dedicate 1.5 hours for each worksheet, 1.5 hours x 4 worksheets = 6 hours total. Angelo and Melanie need to start with planning 12 hours to study, at 4 hours a day, 12 / 4 = 3 days. However, they need to include time for breaks and lunch. Every hour they want to include a 10-minute break, so 12 total hours x 10 minutes = 120 extra minutes for breaks. They also want to include 3 10-minute snack breaks, 3 x 10 minutes = 30 minutes. And they want to include 30 minutes for lunch each day, so 120 minutes for breaks + 30 minutes for snack breaks + 30 minutes for lunch = 180 minutes, or 180 / 60 minutes per hour = 3 extra hours. So Angelo and Melanie want to plan 12 hours to study + 3 hours of breaks = 15 hours total. They want to study no more than 4 hours each day, 15 hours / 4 hours each day = 3.75 They will need to plan to study 4 days to allow for all the time they need. The answer is 4
Question: Mark's basketball team scores 25 2 pointers, 8 3 pointers and 10 free throws. Their opponents score double the 2 pointers but half the 3 pointers and free throws. What's the total number of points scored by both teams added together? Let's think step by step Mark's team scores 25 2 pointers, meaning they scored 25*2= 50 points in 2 pointers. His team also scores 6 3 pointers, meaning they scored 8*3= 24 points in 3 pointers They scored 10 free throws, and free throws count as one point so they scored 10*1=10 points in free throws. All together his team scored 50+24+10= 84 points Mark's opponents scored double his team's number of 2 pointers, meaning they scored 50*2=100 points in 2 pointers. His opponents scored half his team's number of 3 pointers, meaning they scored 24/2= 12 points in 3 pointers. They also scored half Mark's team's points in free throws, meaning they scored 10/2=5 points in free throws. All together Mark's opponents scored 100+12+5=117 points The total score for the game is both team's scores added together, so it is 84+117=201 points The answer is 201 ... Therefore, 1000 - 480 = 520 do not like to play basketball. The percentage of the school that do not like to play basketball is 520/1000 * 100 = 52 The answer is 52

**Tree of Thoughts Prompt Example**

Here's a math word problem: [input]
Current solution steps:[partial solution]
What should be the next step in solving this problem?
Here's a math word problem: [input]
Partial solution: [partial solution]
How likely is this partial solution to lead to the correct answer? (impossible/unlikely/likely/very likely/certain)
Here's a math word problem: [input]
Proposed final answer: [answer]
How likely is this answer to be correct? (impossible/unlikely/likely/very likely/certain)

**Graph of Thoughts Prompt Example**

<Instruction> Merge the following 2 sorted lists of length length1 each, into one sorted list of length length2 using a merge sort style approach. Only output the final merged list without any additional text or thoughts!:</Instruction>
<Approach> To merge the two lists in a merge-sort style approach, follow these steps: 1. Compare the first element of both lists. 2. Append the smaller element to the merged list and move to the next element in the list from which the smaller element came. 3. Repeat steps 1 and 2 until one of the lists is empty. 4. Append the remaining elements of the non-empty list to the merged list. </Approach>
Merge the following two lists into one sorted list: 1: input1 2: input2
Merged list:

**Self-Consistency Prompt Example**

You will be provided with the answer to a question. The question and options are delimited by triple backticks, and the answer is delimited by triple hashtags. Extract the final answer from the provided solution. Return only the letter corresponding to the chosen option (A, B, C, D, or E), prefixed by 'Final answer:'

**LLM Multi-Agent Debate Prompt Example**

Using the solutions from other agents as additional information, can you provide your answer to the math problem? The original math problem is []. Your final answer should be a single numerical number, in the form [answer], at the end of your response.

# D   Comprehensive Results

The following table provides data comparing the performance of each method on the selected benchmarks with different base models.

| | Benchmark | Results | | | | |
|---|---|---|---|---|---|---|
| | | GPT-3.5-turbo | GPT-4o | Claude-3.5-Sonnet | Mixtral 8x22B | Llama-3.1-8B |
| **Unaided** | AQuA | 0.639 | 0.820 | 0.857 | 0.662 | 0.613 |
| | Merging | 0.702 | 0.858 | 0.858 | 0.740 | 0.600 |
| | GSM-Symbolic | 0.733 | 0.893 | 0.847 | 0.740 | **0.580** |
| | GSM8K | 0.767 | 0.913 | 0.963 | 0.707 | 0.593 |
| | LegalBench | 0.673 | 0.687 | 0.620 | 0.670 | 0.587 |
| | MMLU | 0.630 | 0.760 | 0.840 | 0.760 | 0.667 |
| | MedQA | 0.640 | 0.893 | 0.893 | 0.533 | 0.707 |
| | SVAMP | 0.720 | 0.890 | 0.840 | 0.790 | 0.770 |
| | Sorting 032 | 0.961 | 0.992 | 0.985 | 0.968 | 0.818 |
| | TruthfulQA | 0.848 | 0.949 | 0.993 | 0.931 | 0.911 |
| **Chain of Thought** | AQuA | 0.740 | 0.860 | 0.920 | 0.860 | 0.760 |
| | Merging | 0.711 | 0.825 | 0.862 | 0.766 | 0.600 |
| | GSM-Symbolic | 0.753 | 0.920 | 0.940 | 0.853 | 0.764 |
| | GSM8K | 0.700 | 0.960 | 1.000 | 0.900 | 0.745 |
| | LegalBench | 0.620 | 0.680 | 0.747 | 0.653 | 0.620 |
| | MMLU | 0.689 | 0.871 | 0.600 | 0.748 | **0.631** |
| | MedQA | 0.660 | 0.880 | 0.860 | 0.707 | 0.707 |
| | SVAMP | 0.760 | 0.920 | 0.920 | 0.880 | 0.880 |
| | Sorting 032 | 0.960 | 0.992 | 0.951 | 0.768 | 0.850 |
| | TruthfulQA | 0.920 | 0.960 | 0.980 | 0.940 | 0.960 |
| **Self-Consistency** | AQuA | 0.649 | 0.886 | 0.876 | 0.727 | 0.736 |
| | Merging | 0.740 | 0.870 | 0.870 | 0.800 | 0.750 |
| | GSM-Symbolic | 0.804 | 0.972 | 0.957 | **0.463** | **0.520** |
| | GSM8K | 0.700 | 0.902 | 0.966 | 0.900 | 0.767 |
| | LegalBench | 0.667 | 0.567 | 0.600 | 0.580 | 0.600 |
| | MMLU | 0.795 | 0.867 | 0.845 | 0.844 | 0.800 |
| | MedQA | 0.513 | 0.780 | 0.780 | 0.513 | 0.507 |
| | SVAMP | 0.875 | 0.854 | 0.886 | 0.840 | **0.500** |
| | Sorting 032 | 0.975 | 0.990 | 0.985 | 0.972 | 0.930 |
| | TruthfulQA | 0.680 | 0.900 | 0.920 | 0.840 | 0.780 |
| **ReAct** | AQuA | **0.600** | 0.900 | 0.900 | 0.671 | **0.400** |
| | Merging | 0.735 | 0.870 | 0.870 | 0.760 | 0.620 |
| | GSM-Symbolic | 0.873 | 0.933 | 0.867 | 0.800 | **0.630** |
| | GSM8K | 0.760 | 0.960 | 0.980 | 0.920 | 0.857 |
| | LegalBench | 0.887 | 0.82 | 0.915 | 0.520 | 1.000 |
| | MMLU | 0.700 | 0.880 | 0.800 | 0.640 | **0.480** |
| | MedQA | 0.747 | 0.887 | 0.813 | 0.732 | 0.580 |
| | SVAMP | 0.730 | 0.920 | 0.880 | 0.820 | 0.800 |
| | Sorting 032 | 0.980 | 1.000 | 1.000 | 0.960 | 0.923 |
| | TruthfulQA | 0.780 | 0.880 | 0.980 | 0.900 | 0.900 |

(Continues on next page)

| | Benchmark | Results | | | | |
|---|---|---|---|---|---|---|
| | | GPT-3.5-turbo | GPT-4o | Claude-3.5-Sonnet | Mixtral 8x22B | Llama-3.1-8B |
| **Tree of Thoughts** | AQuA | 0.670 | 0.881 | 0.867 | 0.740 | 0.660 |
| | Merging | 0.814 | 0.898 | 0.881 | 0.873 | 0.774 |
| | GSM-Symbolic | 0.707 | 0.751 | 0.627 | 0.794 | 0.647 |
| | GSM8K | 0.640 | 0.950 | 0.970 | 0.860 | 0.760 |
| | LegalBench | 0.813 | 1.000 | 1.000 | 0.593 | 0.687 |
| | MMLU | 0.769 | **0.653** | 0.867 | **0.640** | 0.720 |
| | MedQA | 0.487 | 0.907 | 0.793 | 0.513 | 0.527 |
| | SVAMP | 0.760 | 0.935 | 0.867 | 0.750 | 0.570 |
| | Sorting 032 | 0.982 | 0.998 | 0.998 | 0.985 | 0.881 |
| | TruthfulQA | 0.940 | 0.960 | 0.990 | 0.980 | 0.940 |
| **Graph of Thoughts** | AQuA | 0.700 | 0.840 | 0.900 | 0.840 | 0.760 |
| | Merging | 0.808 | 0.902 | 0.886 | 0.812 | 0.743 |
| | GSM-Symbolic | 0.733 | **0.813** | 0.813 | 0.813 | 0.793 |
| | GSM8K | 0.780 | 0.960 | 0.980 | 0.900 | 0.800 |
| | LegalBench | 0.707 | **0.407** | 0.500 | 0.553 | 0.660 |
| | MMLU | **0.440** | **0.680** | **0.600** | **0.700** | **0.500** |
| | MedQA | 0.727 | 0.913 | 0.818 | 0.838 | 0.873 |
| | SVAMP | 0.800 | 0.920 | 0.940 | 0.900 | 0.820 |
| | Sorting 032 | 0.993 | 0.998 | 0.997 | 0.990 | 0.878 |
| | TruthfulQA | 0.960 | 0.960 | 0.980 | 0.960 | 0.980 |
| **Multi-Agent Debate** | AQuA | 0.680 | 0.960 | 0.780 | 0.800 | 0.640 |
| | Merging | 0.780 | 0.890 | 0.870 | 0.830 | 0.785 |
| | GSM-Symbolic | 0.833 | 0.823 | 0.971 | 0.901 | **0.650** |
| | GSM8K | 0.810 | 0.920 | 0.904 | 0.780 | 0.610 |
| | LegalBench | 0.586 | 0.586 | 0.585 | 0.585 | 0.606 |
| | MMLU | 0.720 | 0.880 | 0.854 | 0.625 | 0.540 |
| | MedQA | **0.353** | 0.907 | 0.836 | 0.558 | 0.636 |
| | SVAMP | 0.902 | 0.958 | 0.916 | 0.833 | 0.800 |
| | Sorting 032 | 0.989 | 0.998 | 0.997 | 0.990 | 0.960 |
| | TruthfulQA | 0.800 | 0.880 | 0.900 | 0.920 | 0.880 |

