# OpenReview forum: "Pitfalls in Evaluating Inference-time Methods for Improving LLM Reliability"
_TMLR — Accepted by TMLR_

### Review · Reviewer_bvGn · 2025-03-11

**Summary Of Contributions:**

This work presents a survey and a comprehensive reproduction study on various inference-time strategies for LLMs. The first part ("Evaluations in the Literature") analyzes the datasets and models evaluated in papers citing Chain-of-Thought over the past three years. The second part ("Experiments" and "Results") systematically evaluates six different inference-time strategies across five LLMs on eight datasets. The authors highlight significant performance variations across different tasks, providing valuable insights into the effectiveness of these strategies.

**Audience:**

Yes

**Broader Impact Concerns:**

I don't have any concerns on it.

**Claims And Evidence:**

Yes

**Requested Changes:**

The current version of the paper lacks sufficient technical contributions to stand out as either a survey or an analysis paper, as both the (1) literature analysis and (2) experimental contributions are relatively incremental compared to existing studies. At a higher level, I recommend that the authors refine the main focus and contributions of the paper. Conducting an in-depth literature analysis or an extensive reproduction study are both valuable directions, but each requires a stronger, more cohesive execution—potentially warranting separate papers.

More concretely:

- For the literature analysis: I recommend expanding the scope by starting from multiple foundational papers rather than relying solely on CoT. Additionally, incorporating a broader analysis—such as tracking the evolution of popular inference-time methods rather than just models and datasets—could provide more meaningful insights. Some methodological aspects, such as how model names are standardized, should also be revisited for accuracy and consistency.

- For the experimental study: The cost analysis is an interesting aspect, but the methodology could be more rigorous. Currently, cost estimations are based on a single model, whereas Table 2 presents results averaged over different models. Revisiting the methodology for cost estimation could improve the experimental rigor. More broadly, while the experiments reveal some interesting findings, they feel scattered in their current form, and the main takeaways are unclear. A stronger narrative tying the results together would significantly improve the impact of this section.

Overall, the paper has potential, but the main message remains unclear. The authors should decide whether this is primarily a reproduction paper or a survey and refine the structure accordingly.

**Strengths And Weaknesses:**

### Strengths

- **Comprehensive Reproduction Study**: The paper provides a thorough reproduction study across multiple models, inference-time methods, and datasets, offering valuable empirical insights.

- **Interesting Dataset and Model Popularity Analysis**: The analysis of datasets and models used in papers citing Chain-of-Thought (CoT) over three years is particularly intriguing. While dataset popularity remains relatively stable, model preferences have shifted significantly over time. (I have minor questions about this—see below.)

### Weaknesses
- **Limited Technical Contributions**: The primary weakness of this work is the lack of significant technical contributions. The main contributions are:

1. A literature analysis of inference-time methods.
2. A reproduction study of representative methods.

However, as the authors acknowledge, several existing survey papers (e.g., [Welleck et al., 2024](https://arxiv.org/abs/2406.16838); [Chu et al., 2024](https://arxiv.org/abs/2309.15402)) already provide in-depth discussions on inference-time methodologies. Additionally, analyzing the datasets and models used in papers citing CoT does not seem to yield particularly novel insights.

Regarding the reproduction experiments, prior work such as [Sprague et al., (2024)](https://arxiv.org/abs/2409.12183), has already demonstrated that the effectiveness of these inference-time strategies depends heavily on the underlying model and dataset. Given this, the paper does not offer sufficiently new empirical findings or an especially in-depth survey to justify its contributions.

I acknowledge the differences between this work and [Welleck et al., 2024](https://arxiv.org/abs/2406.16838) in the survey component and from [Sprague et al., (2024)](https://arxiv.org/abs/2409.12183) in the experiment parts, as discussed in related work. However, despite these distinctions, I do not find that either contribution—(1) the literature analysis of inference-time methods or (2) the reproduction study—offers sufficiently novel insights to justify a strong contribution.

- **Potential Issues with Survey Methodology in Section 3**: I have several major concerns about the analysis presented in Section 3:

(a) Choice of the Starting Paper: Why did the authors choose the CoT paper as the sole starting point? While it is a seminal work, there were concurrent studies at the time. Unless the paper is explicitly positioned as a survey on Chain-of-Thought reasoning papers (as in [Chu et al., 2024](https://arxiv.org/abs/2309.15402)), restricting the analysis to papers citing CoT seems problematic.

(b) Figure 2 Inconsistencies: The categorization in Figure 2 raises some concerns. For example, why is Llama 2-7B-Chat separated from Llama 2-7B, while Llama 3 and Llama models do not distinguish between their chat and base versions?

(c) Possible Data Extraction Errors: The presence of papers mentioning GPT-4o in 2023 is concerning, as this model was released in May 2024. This suggests potential issues in extracting methodologies or misattributed citations.
Overall, while the study presents useful reproduction results and an interesting historical analysis, its limited novelty and potential methodological issues reduce its impact.

---

### Review · Reviewer_p92h · 2025-03-11

**Summary Of Contributions:**

This paper provides a critical analysis of inference-time methods designed to improve the reliability of large language models (LLMs). The authors conduct a thorough survey of evaluation approaches in the existing literature and identify key inconsistencies in benchmark selection, model choices, and experimental methodologies.
To assess the generalizability of claims made in prior research, the paper presents a systematic empirical study evaluating multiple inference-time methods across a diverse range of models and benchmarks. The findings highlight that:
1. Performance improvements reported in prior works are often fragile and do not consistently generalize to different models and tasks.
2. Methods that demonstrate significant improvements on weaker base models may fail to enhance or even degrade performance when applied to stronger models.
3. Benchmark selection plays a crucial role in evaluating inference-time methods, and no single benchmark is representative enough to provide generalizable conclusions.

Overall, this study provides valuable insights into the pitfalls of inference-time reliability evaluations and offers recommendations for more rigorous evaluation practices.

**Audience:**

Yes

**Broader Impact Concerns:**

The paper does not raise ethical concerns.

**Claims And Evidence:**

Yes

**Requested Changes:**

- Provide more detailed theoretical insights (which is critical to securing my recommendation for acceptance): a deeper theoretical discussion on why certain methods exhibit fragile improvements may add depth to the study.
- Include ablation studies on the components of inference-time methods (which would strengthen the work in my view): analyzing the impact of individual components / key parameters within inference-time methods would help clarify which aspects are most responsible for performance variations.
- Expand benchmark selection (which would strengthen the work in my view): Including additional benchmarks for more domain-specific tasks (e.g., medical or legal) would strengthen the generalizability of findings.

**Strengths And Weaknesses:**

- Strengths
  - The paper provides a well-structured survey of inference-time methods and their evaluation strategies, effectively highlighting inconsistencies in prior research.
  - The authors conduct a detailed set of experiments across multiple LLMs and benchmarks, ensuring a broad and representative analysis.
  - The results emphasize critical challenges in reproducibility and generalizability of inference-time reliability improvements, offering important contributions to the field.
- Weaknesses
  - While the empirical findings are strong, the paper lacks a deeper theoretical discussion on why certain methods fail to generalize across different models and tasks.
  - The paper does not systematically analyze the impact of specific components within each inference-time method, which could help identify key factors affecting performance.
  - While the study evaluates a diverse set of benchmarks, the selection is still relatively narrow given the vast number of tasks LLMs can handle. More benchmarks from domains such as medical, legal, or multilingual settings could strengthen the study.

---

### Review · Reviewer_sSPU · 2025-03-17

**Summary Of Contributions:**

This paper surveyed existing inference-time methods designed to improve the reliability of Large Language Model (LLM) outputs, highlighting substantial variability in benchmarks and models used in prior evaluations. The authors conducted their own experiments, revealing that while these methods can enhance reliability, their effectiveness varies significantly across tasks and domains. Furthermore, methods successful on weaker base models often fail to show improvements when applied to stronger models.

**Audience:**

Yes

**Broader Impact Concerns:**

I don't have ethical concerns about this work.

**Claims And Evidence:**

Yes

**Requested Changes:**

please see the weaknesses above.

**Strengths And Weaknesses:**

strengths:
* The paper presents a comprehensive analysis of various inference-time methods, highlighting their effectiveness across different LMs and benchmarks.


weakness:
* The paper acknowledges significant variability across benchmarks and models but lacks clear guidelines or actionable recommendations for future research to address this variability.
* Discussion on computational costs is brief and does not adequately address trade-offs between cost and effectiveness, leaving practical applicability concerns insufficiently explored.
* The reproducibility analysis, though useful, is limited to GPT-3.5-turbo, restricting insight into reproducibility issues for other widely-used models.

---

### Author Response · Authors · 2025-03-22
**Responses to Reviewer Comments**

## Responses to Reviewer sSPU

### Comment 1

> The paper acknowledges significant variability across benchmarks and models but lacks clear guidelines or actionable recommendations for future research to address this variability.

**Response:** Our findings highlight the need for more robust evaluations to address variability across models and benchmarks. Although we make several recommendations implicitly throughout our paper, we agree with the reviewer that we should call these out more explicitly. Our specific actionable recommendations are:

* **Evaluate Across Diverse Models:**
  Methods that improve weaker models may underperform or degrade results with stronger models. Work on inference-time methods should assess methods on a range of models, including state-of-the-art systems.
* **Select a Representative Suite of Benchmarks:**
  No single benchmark reliably captures the performance of an inference-time method. Researchers should curate diverse benchmarks that cover various reasoning types and task formats. We do not think it is clear enough yet to recommend a specific suite of benchmarks, but we encourage the community to work towards developing such a benchmark suite and demonstrating its utility in evaluation.
* **Prioritize Benchmark Diversity Over Quantity:**
  Evaluating a narrow set of benchmarks risks overfitting to specific tasks. A diverse selection provides a more balanced assessment of method effectiveness.
* **Adopt Standardized Evaluation Protocols:**
  The wide variation in evaluation methods underscores the need for standardized protocols to enhance comparability and reliability across studies. We recommend that researchers at a minimum always include results from the more widely used benchmarks, along with additional benchmarks that they make a case for using for evaluating their work.

### Comment 2

> The reproducibility analysis, though useful, is limited to GPT-3.5-turbo, restricting insight into reproducibility issues for other widely-used models.

**Response:** We understand that this is commenting on Section 5.2, which only considered GPT-3.5-turbo in analyzing reproducibility and unlike other parts of our evaluation where we use several models. Our decision to only use GPT-3.5-turbo for the reproducibility analysis was driven by practical necessity rather than preference:

* **Limited Viable Alternatives:** After reviewing the landscape of available models suitable for our reproducibility analysis, GPT-3.5-turbo emerged as effectively the only viable option. Most models used in the original papers are either deprecated, inaccessible, or unstable for long-term research.

* **Model Deprecation Reality:** The majority of models used in the original papers, such as text-ada-001, text-babbage-001, code-davinci-001, and LaMDA, have been deprecated or are no longer publicly available. Even PaLM, though technically available until April 2025, is about to be deprecated.

* **Standardization Requirement:** Using GPT-3.5-turbo provided a consistent baseline across all methods.

* **Research Community Adoption:** Use of GPT-3.5-turbo has become standard for implementing these techniques in current research, including its adoption by resources like the Chain of Thought Hub.

Our decision to use GPT-3.5-turbo for reproducibility analysis was necessitated by OpenAI's deprecation schedule. As documented by OpenAI, all legacy models including text-davinci-002 and text-davinci-003 were officially shut down on January 4, 2024, with GPT-3.5-turbo-instruct designated as the recommended replacement [https://platform.openai.com/docs/deprecations](https://platform.openai.com/docs/deprecations). GPT-3.5-turbo emerged as the most viable option that balances availability, capability, and consistency across methods.

To illustrate the limited model availability, below is a table showing the models used in the original papers for each method:

| Method | Original Models Evaluated in Papers |
|--------|---------------------|
| Chain of Thought | GPT-3 (text-ada-001, text-babbage-001, text-curie-001, text-davinci-002), LaMDA, PaLM, UL2 20B, Codex |
| Self-Consistency | UL2, GPT-3 (code-davinci-001, code-davinci-002), LaMDA-137B, PaLM-540B |
| ReACT | PaLM-540B, GPT-3 (text-davinci-002) |
| Tree of Thought | GPT-4, GPT-3.5 |
| Graph of Thought | GPT-3.5 |
| Debate | GPT-3.5 |

As this table demonstrates, GPT-3.5-turbo is the only model that offers both the necessary capabilities and long-term availability for reproducible research across all these methods. The older models are largely deprecated, while newer alternatives either don't support all the methods or are too costly for large experimentation.

---

### Author Response · Authors · 2025-03-22
**Responses to Reviewer Comments**

## Responses to Reviewer p92h

### Comment 1

> "Provide more detailed theoretical insights (which is critical to securing my recommendation for acceptance): a deeper theoretical discussion on why certain methods exhibit fragile improvements may add depth to the study."

**Response:** We appreciate the reviewer's interest in theoretical results, but our work is fundamentally empirical so it is hard to understand what kind of theory could fit with this work. Our paper is an empirical evaluation of how inference-time methods perform across diverse models and benchmarks to identify patterns in their effectiveness. This kind of empirical study we think is well motivated by the current state of the research community, and that there is strong interest and value in both understanding how to interpret and improving the quality of empirical results in this area. We completely agree with the reviewer that the lack of any theoretical basis for most work in this area is serious (as one example, the Chain-of-Thought paper does not include a single definition, theorem, or proof, as is the case for the overwhelming majority of papers in our literature review), but we would argue that the lack of any meaningful theory in this area amplifies the importance of doing empirical evaluations well and emphasizes the need for our study.

### Comment 2

> "Include ablation studies on the components of inference-time methods (which would strengthen the work in my view): analyzing the impact of individual components / key parameters within inference-time methods would help clarify which aspects are most responsible for performance variations."

**Response:** We agree that ablation studies are important in evaluating a specific method, but since our goal is to compare evaluations of different methods that include different components, it is not clear to us what kind of ablation study would be helpful. The methods we evaluate all have different components and different hyperparameters, so it is not possible to compare their impact across the methods. Instead, our analysis evaluates multiple inference-time methods applied across a diverse set of models and benchmarks to assess their overall effectiveness and variability. Performing ablation studies on individual components of specific methods would not align with the primary objective of our work, which is to provide a broad, comparative assessment rather than an in-depth exploration of parameter-level contributions within individual methods.

### Comment 3

> "Expand benchmark selection (which would strengthen the work in my view): Including additional benchmarks for more domain-specific tasks (e.g., medical or legal) would strengthen the generalizability of findings."

**Response:** We agree with this comment. We will expand our evaluation to include additional domain-specific benchmarks such as MEDQA for medical reasoning and LegalBench for legal reasoning. We are working on adding these additions and hope to have results before the end of the response period.

---

### Author Response · Authors · 2025-03-22
**Responses to Reviewer Comments**

## Responses to Reviewer bvGn

### Comment 1

> "For the literature analysis: I recommend expanding the scope by starting from multiple foundational papers rather than relying solely on CoT. Additionally, incorporating a broader analysis—such as tracking the evolution of popular inference-time methods rather than just models and datasets—could provide more meaningful insights. Some methodological aspects, such as how model names are standardized, should also be revisited for accuracy and consistency."

**Response:**
Regarding the comment to expand the scope of the literature analysis, we conducted a broader citation analysis by examining the overlap between the citation chains of other foundational papers in the field for the reviewer's consideration. This analysis goes beyond the original Chain-of-Thought paper and includes other works on inference-time methods and model alignment. Since specific seminal papers were not mentioned in the reviewer's comment, we chose four that we thought would offer insight. Below are the results of the citation chain overlap with the Chain-of-Thought paper:

**Table 1: Paper Citation Counts**

| Paper | Citations |
|-------|-----------|
| Chain-of-Thought Prompting Elicits Reasoning in Large Language Models | 7,312 |
| Retrieval-Augmented Generation for Knowledge-Intensive NLP Tasks | 4,829 |
| Self-Instruct: Aligning Language Models with Self-Generated Instructions | 1,896 |
| Large Language Models Can Self-Improve | 511 |
| Solving Quantitative Reasoning Problems with Language Models | 667 |

**Table 2: Citation Overlap with Chain-of-Thought Paper**

| Paper | Overlap Count | Overlap Percentage |
|-------|---------------|-------------------|
| Retrieval-Augmented Generation | 774 | 16.03% |
| Self-Instruct | 377 | 19.88% |
| Large Language Models Can Self-Improve | 197 | 38.55% |
| Solving Quantitative Reasoning Problems | 309 | 46.33% |

The overlap suggests that focusing on Chain-of-Thought's citation network captures a representative sample of the literature on inference-time methods. This overlap indicates that many researchers are building on multiple foundational works simultaneously.

Regarding the methodological concerns about model name standardization, we will implement a more rigorous normalization process for our literature analysis, including consistent version numbering for models and updated timeline extraction. We appreciate this feedback and agree it will enhance the accuracy and consistency of our analysis.

### Comment 2

> "For the experimental study: The cost analysis is an interesting aspect, but the methodology could be more rigorous. Currently, cost estimations are based on a single model, whereas Table 2 presents results averaged over different models. Revisiting the methodology for cost estimation could improve the experimental rigor. More broadly, while the experiments reveal some interesting findings, they feel scattered in their current form, and the main takeaways are unclear. A stronger narrative tying the results together would significantly improve the impact of this section."

We understand the reviewer's concern about the apparent methodological inconsistency in our cost analysis. To clarify, our primary interest was measuring how cost depends on the inference-time method itself, quantified by the number of API calls to the underlying model. We made the assumption that this cost structure would not vary significantly across different models for the same method, which is why we initially reported costs only for GPT-3.5-turbo.

To test this assumption, we ran a preliminary cost experiment with ReAct and all else being equal as the underlying assumption. However, instead of running the full experiments again, we processed 450 problems per method and model. The results for GPT-3.5-turbo are the same as what was reported in the paper. The additional results are:

| Model | CoT | SC | ReAct | ToT | GoT | Debate |
|-------------|-----|-----|-------|------|------|--------|
| GPT-3.5-turbo | 3.5 | 10.0 | 5.2 | 53.8 | 37.2 | 6.0 |
| GPT-4o | 3.5 | 10.0 | 2.07 | 53.8 | 37.2 | 6.0 |
| Claude 3.5-Sonnet | 3.5 | 10.0 | 2.50 | 53.8 | 37.2 | 6.0 |
| Llama-3.1-8B-Instruct | 3.5 | 10.0 | 6.76 | 53.8 | 37.2 | 6.0 |
| Mixtral 8x22B | 3.5 | 10.0 |4.21 | 53.8 | 37.2 | 6.0 |

---

### Decision · Action_Editor_5L9N · 2025-05-04

**Recommendation:** Accept with minor revision

**Comment:**

This paper investigates the replicability and generalizability of the inference-time techniques for improving the reliability of LLMs. Through their own experiments on six different inference-time strategies for five LLMs on eight datasets, the authors demonstrate that although these techniques can boost reliability, their effectiveness differs greatly depending on the task and domain, and approaches that work well with less capable models often do not yield the same benefits on more advanced ones.

Overall, this is a valuable study with substantial and clear empirical evidence to support the claim, and it provides insights and guidance that would benefit researchers in this area.

Strengths
- The paper provides an in-depth, comprehensive reproducibility evaluation of multiple inference-time techniques, and it provides insights on how well they perform across a range of language models and benchmarks (Reviewer sSPU,  Reviewer p92h, Reviewer bvGn).
- The dataset and model popularity analysis is particularly interesting (Reviewer bvGn).
- This paper is overall well written (Reviewer sSPU,  Reviewer p92h).

There are several revisions to incorporate as follows.
- Improve the organization of the paper by building a stronger connection between the two parts: (1) the literature analysis and (2) the reproduction study. For example, the author can explain how the findings of the literature analysis can inform the reproduction study (Reviewer bvGn).
- Provide the paper citation analysis and a more rigorous model name standardization to enhance the accuracy and consistency of literature analysis (Reviewer bvGn).
- Provide a clear discussion to distinguish from other existing work such as Welleck et al., 2024; Chu et al., 2024; Sprague et al., 2024 (Reviewer bvGn).
- Explain the choice of GPT-3.5 to justify the design of experiments for reproducibility and cost analysis (Reviewer bvGn, Reviewer sSPU).
- Add the actionable recommendations to address the limitation of performance variation. The author has acknowledged this and provided several recommendations in their response (Reviewer sSPU).
- Add more results to expand the evaluation on other domains, such as medical and legal (Reviewer p92h).

**Audience:**

Yes

**Claims And Evidence:**

Yes